# Safe stool disposal and associated factors among mothers of children aged under-two years in Gambia: Evidence from Gambia Demographic Health Survey 2019/20

**Menen Tsegaw**[1]*, **Bezawit Mulat**[2], **Kegnie Shitu**[3]

1 Department of Public Health, College of Medicine and Health Sciences, Ambo University, Ambo, Ethiopia, 2 Department of Human Physiology, School of Medicine, College of Medicine and Health Sciences, University of Gondar, Gondar, Ethiopia, 3 Department of Health Education and Behavioral Sciences, Institute of Public Health, College of Medicine and Health Sciences, University of Gondar, Gondar, Ethiopia

* menentsegaw@gmail.com

## Abstract

### Background

Appropriate disposal of child stool is vital in preventing the spread of faecal-oral diseases. According to WHO/ UNICEF Joint Monitoring Program, Safe child stool disposal includes disposing a stool in a Flush or pour-flush toilet/latrine (to a piped sewer system, septic tank, pit latrine), Ventilated improved pit (VIP) latrine or a Pit latrine with slab.

### Objective

The study aimed to assess safe child stool disposal practice and associated factors among mothers with children aged under-two years in Gambia.

### Methods

This study was based on a large community-based cross-sectional survey, conducted from 21 November 2019 to 30 March 2020 in Gambia. The survey employed a two-staged stratified cluster sampling technique to recruit study participants. Descriptive statistics and logistic regression models were used to summarize descriptive data and identify factors associated with safe waste disposal, respectively. A p-value of less than 0.05 and 95% confidence interval were used to determine statistical significance.

### Results

The prevalence of safe stool disposal among mothers with children aged under-two years were 56.3% (95% CI: 54.6%, 58.1%). Mothers aged 25–34 (AOR = 0.78 (95%CI: 0.62, 0.98)), the highest wealth quintile (AOR = 0.43 (95%CI: 0.33, 0.56)), being exposed to media (AOR = 1.37 (95%CI: 1.07, 1.76)), increasing age of children (AOR = 1.06 (1.05, 1.07)), Being employed (AOR = 1.31 (1.11, 1.55)) and Geographic region were significantly associated with safe child disposal practice.

**Data Availability Statement:** https://dhsprogram.com/data/dataset/Gambia_Standard-DHS_2019.cfm.

**Funding:** The author(s) received no specific funding for this work.

**Competing interests:** The authors have declared that no competing interests exist.

**Abbreviations:** COR, (Crude Odds Ratio); AOR, (Adjusted Odds Ratio); CI, Confidence interval; GDHS, Gambia Demographic Health Survey.

## Conclusion

The prevalence of safe child stool disposal was low in Gambia. Age of the mother, age of the child, region, wealth index, media exposure and occupational status of the mother were significantly associated with safe child stool disposal. Public health intervention strategies designed to promote safe child stools disposal need to conduct thorough community assessments to identify community-specific facilitators, needs and barriers. Additionally, public health experts and policy makers should take into consideration the geographical and wealth inequalities when designing programs aimed to improve safe child stool disposal practice.

## Introduction

According to WHO/ UNICEF Joint Monitoring Program, Safe child stool disposal is defined as disposing stools in one of the following options: Flush or pour-flush toilet/latrine (to a piped sewer system, septic tank, pit latrine), Ventilated improved pit (VIP) latrine or a Pit latrine [1, 2]. Appropriate disposal of child stool is vital in preventing the spread of faecal-oral diseases [3–5]. Sustainable development Goal (SDG) 6 aims to "achieve access to adequate and equitable sanitation and hygiene for all and end open defecation, paying special attention to the needs of women and girls and those in vulnerable by 2030 [6]. Achieving the widespread scope of SDG by 2030 needs quadrupling current rates of advance in securely overseen drinking water administrations, sanitation administrations and basic hygiene services especially in least developing countries [7]. Elimination of open defecation is one of the core priority solutions to reduce global disparities in Water, Sanitation, and Hygiene (WASH) [8].

Different studies revealed that unsafe child stool disposal was one of the major factors that causes enteric infections including Diarrheal diseases [3, 9–13]. Diarrhea is one of the most common causes of under-five mortality rate [14–16]. There is a common misunderstanding that child faeces are less infectious than adult [17, 18]. Besides the Water and Sanitation Project implemented by the Department of Water Resources, to introduce hygienic means of excreta disposal in the entire country, there has not been much-coordinated policy response to basic sanitation issues in Gambia [19]. Different studies have been done on safe child stool disposal and associated factors among children but there is no study done in Gambia. Among the studies that investigated child feces disposal, most found that the maternal age, higher maternal education, highest household wealth index, availability of toilet facility, and improved drinking water were significantly and positively associated safe child stool disposal [20–22]. According to the Gambia Demographic Health Survey (GDHS) 2019/20, safe child stool disposal among children under the age of 2 years has been decreased from 82% in 2013 to 56% in 2019–20. Although the magnitude of safe child stool disposal is known the possible factors affecting safe child stool disposal have not yet been well studied. Therefore, the current study is aimed to assess factors that are significantly associated with safe child stool disposal practice in Gambia using the GDHS 2019–20.

## Methods

### Study design and setting

This study was based on a large community-based survey, Gambia Demographic Health Survey (GDHS), conducted from 21 November 2019 to 30 March 2020 in Gambia. Gambia is

located on the West African coast. It is bordered on the North, South and East by the Republic of Senegal and on the West by the Atlantic Ocean. The survey employed a stratified two-stage cluster sampling. In the first stage, Enumeration Areas (EAs) were selected with a probability proportional to their size within each sampling stratum. In the second stage, the households were systematically sampled. A total of 3,011, weighted sample of mothers who have child/children under the age of two years were included in the study.

## Variables of the study

**Dependent variable.** Safe child stool disposal was the outcome variable of this study which was obtained after categorizing child stool disposal in to two as safe (Flush or pour-flush toilet/latrine to (piped sewer system, septic tank, pit latrine), Ventilated improved pit (VIP) latrine and Pit latrine with slab) and unsafe (pit latrines without a slab or platform, hanging latrines and bucket latrines).

**Independent variables.** Age of the mother, religion, region, educational level, occupation, residence, wealth index, media exposure (yes, no), age of the child, toilet facility (improved, unimproved), sex of the child, birth order, visited a health facility in the last 12 months (yes, no), history of diarrhea in the last 2 weeks preceding the survey (yes, no), source of drinking water (improved, unimproved) and the number of living children were included. Those variables were identified after reviewing different literature [5, 17, 20, 21, 23].

**Data analysis.** The mother's individual sample weightings were used in the estimation to provide nationally representative results. Descriptive studies like frequency count and proportion for categorical data were used to summarize descriptive data. Bivariable logistic regression was used to select candidate variables for multivariable logistic regression. In the Bivariable logistic regression, a p-value<0.2 was used as a cut of point to select variables for the multivariable analysis entry. Multivariable logistic regression was used to identify independent predictors of safe child stool disposal in Gambia. A 95% confidence interval (CI) and p-value<0.05 were used to determine the statistical significance. Stata version 14.0 was used for statistical analysis.

## Results

The mean age of the mothers was 28.35 with a standard deviation of 6.65 and range of 15–49 years. Regarding religion, 2,952 (98.0%) of mothers were Muslims. About 1,946 (64.6%) of the mothers were from urban, 1202 (39.9%) were from Brikama region, 1,337 (44.4%) of mothers have no education, 1177 (39.1) of mothers were employed and 1314 (43.6%) of mothers were from the poor families (Table 1).

### Prevalence of safe children's stools disposal

The prevalence of safe stool disposal among mothers whose child aged under two years was 56.3% (95%CI: 54.5, 58.1%) (Table 2).

### Factors associated with safe children's stool disposal

In the Bivariable logistic regression age of the mother, religion, region, educational level, occupation, residence, wealth index, media exposure, age of the child, toilet facility, birth order, visited health facility in the last 12 months, and the number of living children, sex of the child, diarrhea history in the last 12months and source of drinking water were independently fitted to the outcome variable. However, only age of the mother, religion, region, educational level, occupation, residence, wealth index, media exposure, age of the child, toilet facility, birth

**Table 1. Socio-demographic characteristics by waste disposal status among mothers/caregivers of children with the age < 2 years in Gambia, 2019/20 (n = 3011).**

| Variables | Child waste disposal practice | | | | Total | |
|---|---|---|---|---|---|---|
| | Unsafe | | Safe | | | |
| | Frequency | Percent | frequency | Percent | Frequency | Percent |
| Age of the mother | | | | | | |
| 15–24 | 367 | 43.8 | 471 | 56.2 | 838 | 27.8 |
| 25–34 | 700 | 44.5 | 874 | 55.5 | 1574 | 52.3 |
| >34 | 246 | 41.1 | 353 | 58.9 | 599 | 19.9 |
| Region | | | | | | |
| Banjul | 17 | 66.9 | 8 | 33.1 | 26 | 0.9 |
| Kaniking | 348 | 68.7 | 159 | 31.3 | 507 | 16.9 |
| Brikama | 559 | 46.5 | 643 | 53.5 | 1202 | 39.9 |
| Mansakonko | 23 | 18.1 | 106 | 81.9 | 129 | 4.3 |
| Kerewan | 153 | 41.2 | 219 | 58.8 | 372 | 12.3 |
| Kuntaur | 78 | 41.1 | 111 | 58.9 | 189 | 6.3 |
| Janjanbureh | 40 | 21.0 | 152 | 79.0 | 193 | 6.4 |
| asse | 95 | 24.1 | 299 | 75.9 | 394 | 13.0 |
| Mothers educational level | | | | | | |
| No education | 514 | 38.5 | 822 | 61.5 | 1337 | 44.4 |
| Primary | 225 | 39.3 | 348 | 60.7 | 574 | 19.1 |
| Secondary | 502 | 51.4 | 474 | 48.6 | 976 | 32.4 |
| Higher | 72 | 57.5 | 53 | 42.5 | 125 | 4.1 |
| Occupation | | | | | | |
| Employed | 612 | 52.0 | 565 | 48.0 | 1177 | 39.1 |
| Unemployed | 702 | 38.3 | 1132 | 61.7 | 1834 | 60.9 |
| Religion | | | | | | |
| Islam | 1284 | 43.5 | 1668 | 56.5 | 2952 | 98.0 |
| Christian | 30 | 49.6 | 30 | 50.4 | 60 | 2.0 |
| Residence | | | | | | |
| Urban | 996 | 51.2 | 949 | 48.8 | 1946 | 64.6 |
| Rural | 317 | 29.8 | 748 | 70.2 | 1066 | 35.4 |
| Wealth index | | | | | | |
| Poor | 427 | 32.3 | 887 | 52.2 | 1314 | 43.6 |
| Middle | 199 | 15.1 | 430 | 25.3 | 629 | 20.9 |
| Rich | 688 | 52.3 | 380 | 22.5 | 1068 | 35.5 |
| Media exposure | | | | | | |
| No | 91 | 6.9 | 161 | 9.5 | 252 | 9.0 |
| Yes | 1223 | 93.1 | 1536 | 90.5 | 2759 | 91.0 |
| Age of the child | | | | | | |
| 0–5 months | 502 | 56.1 | 394 | 43.9 | 897 | 29.7 |
| 6 to 11 months | 348 | 47.5 | 384 | 52.5 | 732 | 24.3 |
| 12 to 17 months | 289 | 36.7 | 499 | 63.3 | 787 | 26.2 |
| 18 to 23 months | 175 | 29.34 | 421 | 70.66 | 596 | 19.8 |
| Sex of the child | | | | | | |
| Male | 663 | 42.6 | 893 | 57.4 | 1556 | 51.7 |
| Female | 651 | 44.7 | 804 | 55.3 | 1456 | 48.3 |
| Toilet facility | | | | | | |
| Unimproved | 388 | 33.4 | 773 | 66.6 | 1161 | 38.5 |
| Improved | 926 | 50.0 | 925 | 50.0 | 1851 | 61.5 |

*(Continued)*

**Table 1.** (Continued)

| Variables | Child waste disposal practice | | | | Total | |
|---|---|---|---|---|---|---|
| | Unsafe | | Safe | | | |
| | Frequency | Percent | frequency | Percent | Frequency | Percent |
| Source of drinking water | | | | | | |
| Unimproved | 136 | 40.58 | 199 | 59.42 | 334 | 11.10 |
| Improved | 1178 | 44.01 | 1499 | 55.99 | 2677 | 88.90 |
| History of Diarrhea | | | | | | |
| No | 988 | 44.7 | 1223 | 55.3 | 2211 | 73.4 |
| Yes | 326 | 40.7 | 474 | 59.3 | 801 | 26.6 |
| Birth order | | | | | | |
| First | 307 | 49.1 | 318 | 50.9 | 625 | 20.8 |
| Second | 260 | 47.2 | 291 | 52.8 | 552 | 18.3 |
| Third | 241 | 46.2 | 281 | 53.8 | 523 | 17.4 |
| Fourth and above | 505 | 38.5 | 807 | 61.5 | 1312 | 43.5 |
| Visited health facility | | | | | | |
| No | 115 | 4.1 | 128 | 4.9 | 2768 | 9.0 |
| Yes | 1198 | 43.3 | 1569 | 56.7 | 2768 | 91.9 |
| Number of living children | | | | | | |
| 1 child | 318 | 48.0 | 344 | 52.0 | 662 | 22.0 |
| 2 children | 287 | 49.1 | 297 | 50.9 | 584 | 19.4 |
| 3 children | 249 | 45.8 | 295 | 54.2 | 544 | 18.0 |
| 4 and above | 460 | 37.7 | 762 | 62.3 | 1222 | 40.6 |

order, visited health facility in the last 12 months, and the number of living children exhibited an association at p-value less than 0.2 and passed to be entered in to the multivariable logistic regression. In the multivariable logistic regression, age of the mothers, occupational status of the mother, household wealth index, region, age of the child and media exposure were significantly associated with safe child stool disposal at p-value of less than 0.05 (Table 3).

The odds of safe stool disposal was 22% (AOR = 0.78 (95%CI: 0.62, 0.98)) lower among mothers aged 25–34 as compared with mothers aged 15–24 year. Region of mothers were significantly associated with safe stool disposal. The odds of safe stool disposal were 56% (AOR = 0.43, 95%CI: 0.44, 0.88) lower among rich households as compared to those with poor wealth quantiles. The odds of safe stool disposal was 1.31 (AOR = 1.31 (95%CI: 1.11, 1.55)) times higher among employed women as compared with unemployed mothers.Age of the

**Table 2. The percentage of safe stool disposal among mothers with children aged under-two years using GDHS 2019/20 (n = 3011).**

| variables | Weighted frequency | Weighted percent (95% CI) |
|---|---|---|
| Type of toilet facility | | |
| Used toilet/latrine | 1697 | 56.3(54.5, 58.1) |
| Put/rinsed into drain/ditch | 190 | 6.3 (5.4,7.2) |
| Throw into garbage | 1097 | 36.4 (34.7,38.1) |
| Left in the open/not disposed of | 22 | 0.8 (0.4,1.1) |
| Other | 5 | 0.2 (0.0,0.4) |
| Child waste disposal | | |
| Safe | 1697 | 56.3 (54.5, 58.1) |
| Unsafe | 1314 | 43.7 (41.8, 45.4) |

**Table 3. Logistic regression for factors associated with safe stool disposal among mothers with children aged under-two years using GDHS, 2019/20 (n = 3011).**

| Variables | COR(95%CI) | P-value | AOR(95%CI) | p-value |
|---|---|---|---|---|
| Age of the mother | | | | |
| 15–24 | 1 | | 1 | |
| **25–34** | **0.78 (0.79,1.09)** | **0.42** | **0.78 (0.62, 0.98)\*** | **0.033** |
| >34 | 1.16 (0.94,1.42) | 0.17 | 0.75 (0.53, 1.04) | 0.088 |
| Region | | | | |
| Banjul | 1 | | 1 | |
| Kaniking | 0.93 (0.61,1.41) | 0.73 | 1.04 (0.67, 1.62) | 0.869 |
| **Brikama** | **2.44 (1.66,3.570** | **<0.001** | **2.33 (1.55, 3.50)\*** | **<0.001** |
| **Mansakonko** | **9.10 (5.60,14.26)** | **<0.001** | **5.98 (3.60, 9.94)\*** | **<0.001** |
| **Kerewan** | **2.86 (1.93,4.22)** | **<0.001** | **1.90 (1.22, 2.97)\*** | **0.005** |
| **Kuntaur** | **2.70 (1.83,3.98)** | **<0.001** | **1.64 (1.03, 2.61)\*** | **0.037** |
| **Janjanbureh** | **6.60 (4.36,10.01)** | **<0.001** | **4.48 (2.77, 7.25)\*** | **<0.001** |
| **Basse** | **6.75 (4.58,9.94)** | **<0.001** | **5.01 (3.23, 7.78)\*** | **<0.001** |
| Mothers educational level | | | | |
| No education | 1 | | 1 | |
| Primary | 1.04 (0.86, 1.31) | 0.71 | 1.09 (0.89, 1.37) | 0.389 |
| Secondary | 0.61 (0.52, 0.72) | <0.001 | 0.97 (0.79, 1.18) | 0.747 |
| Higher | 0.44 (0.31, 0.61) | <0.001 | 1.06 (0.65, 1.75) | 0.809 |
| Occupation | | | | |
| Unemployed | 1 | | 1 | |
| **Employed** | **1.60 (1.38, 1.85)** | **<0.001** | **1.31 (1.11,1.55)\*** | **0.001** |
| Religion | | | | |
| Islam | 1 | | 1 | |
| Christian | 0.63 (0.34, 1.18) | 0.15 | 1.09 (0.54, 2.20) | 0.801 |
| Residence | | | | |
| Urban | 1 | | | |
| Rural | 2.27 (1.96,2.62) | <0.001 | 1.16 (0.90, 1.48) | 0.249 |
| Wealth index | | | | |
| Poor | 1 | | 1 | |
| Middle | 1.00 (0.83, 1.22) | 0.975 | 1.16(0.92, 1.48) | 0.208 |
| **Rich** | **0.29 (0.24, 0.34)** | **<0.001** | **0.43(0.33, 0.56)\*** | **<0.001** |
| Media exposure | | | | |
| No | 1 | | | |
| **Yes** | **0.87 (0.69, 1.08)** | **0.223** | **1.37 (1.07, 1.76)\*** | **0.014** |
| Age of the child | 1.09 (1.06, 1.13) | 0.001 | 1.06 (1.05, 1.07) | <0.001 |
| Sex of the child | | | | |
| Male | 1 | | | |
| Female | 0.97 (0.84, 1.12) | 0.69 | - | |
| Toilet facility | | | | |
| Unimproved | 1 | | | |
| improved | 0.61 (0.53,0.70) | <0.001 | 1.14 (0.95, 1.36) | 0.150 |
| Source of drinking water | | | | |
| Unimproved | 1 | | | |
| improved | 0.98 (0.78,1.22) | 0.84 | - | |
| Diarrhea in the last two weeks | | | | |
| No | 1 | | | |
| Yes | 1.09 (0.93, 1.28) | 0.27 | - | |

*(Continued)*

**Table 3.** (Continued)

| Variables | COR(95%CI) | P-value | AOR(95%CI) | p-value |
|---|---|---|---|---|
| Birth order | | | | |
| First | 1 | | | |
| Second | 1.08 (0.86,1.36) | 0.49 | 1.06 (0.81, 1.38) | 0.674 |
| Third | 1.08 (0.86,1.36) | 0.49 | 1.07 (0.78, 1.47) | 0.680 |
| Fourth and above | 1.48 (1.23,1.78) | <0.001 | 1.39 (0.93, 2.10) | 0.106 |
| Visited health facility | | | | |
| No | 1 | | | |
| Yes | 1.25 (0.97,1.60) | 0.08 | 1.08 (0.81, 1.42) | 0.600 |
| Number of children born | 1.09 (1.06, 1.13) | <0.001 | 1.01 (0.93, 1.11) | 0.841 |

Note: (-): Not applicable for AOR, (*): statistically significant at p-value<0.05

child was significantly associated with safe stool disposal. As the age of the child increases from 0–2 year the odds of safe stool disposal increases by 6% (AOR = 1.06 (95%CI: 1.05, 1.07).

Media exposure was significantly associated with safe stool disposal. The odds of safe stool disposal were 1.37 (AOR = 1.37 (95%CI: 1.07, 1.76)) times higher among mothers who were exposed to at least one form of media as compared with their counterparts (Table 3).

## Discussions

The study aimed to assess safe child stool disposal and associated factors among mothers having a child under the age of two years. The prevalence of safe stool disposal among mothers whose child was aged below two years of age was 56.3% (95%CI: 54.5%, 58.1%)). The finding of the study is higher than studies done in Eswatini, Ethiopia, Bangladesh, India, and Orissa [3, 17, 22, 24, 25]. On the other hands, the finding of the current study is lower than the study conducted in Malawi, Cambodia and one study which included many countries across Sub-Saharan Africa [21, 26, 27]. The possible reason for the discrepancies might be differences in sample size, sampling technique, method of analysis and differences in population characteristics. The study conducted in Sub-Saharan Africa used large sample size and it incorporates individual and contextual factors of different countries that might affect safe stool disposal practice. contextual variations and commonalities might affect safe disposal of children's faeces [21, 28].

Region of the mother is associated with safe stool disposal. Mothers from Brikama, mansakonko, kerewan, kuntaur, janjanbureh, Basse were more likely to disposed stool safely as compared with those from Banjul region of Gambia. The possible reason might be different regions of a country may have different access to health care service, infrastructure and different level of health literacy.

Safe stool disposal was significantly associated with the age of the mother. Women in the age group of 25–34 years were less likely to dispose child's stool safely as compared with women in the age group of f 15–24 years. The finding is consistent with the studies done in Nigeria [29]. This finding is inconsistent with studies done in Sub-Saharan Africa [21]. The possible reason for the discrepancy might be socio-cultural and socio-demographic differences across countries. The other possible reason might be time variation in which our study used only 2019/20 GDHS data but the study in sub-Saharan Africa used DHS data 2015–2018 in 15 sub-Saharan countries. This means, due to time variation there might be different initiatives, policies, strategies and efforts made to improve stool disposal practice among mothers with children aged under-two years.

Safe stool disposal was associated with the age of children. The odds of safe stool disposal increases with increasing child age. This is finding is consistent with the studies done in Malawi, Ethiopia, Eswatini and Ghana [17, 20, 26, 30]. The possible reason might be that as children get older especially starting from 6 months as they start complementary feeding their stool becomes offensive which increases the likelihood of safe stool disposal in older children as compared to younger children less than 6 month whose stool is not offensive making it favorable to defecate on clothes/diapers. The other possible reason might be that some women perceive that faeces of infants especially less than six months are not that much larger and offensive as compared with those who start complementary feeding, which may in turn increases the possibility of disposing the faeces of those older children safely.

Those from households of highest wealth quantiles were less likely to dispose of stool as compared with those from the poor wealth quantiles. Although many studies have revealed that, the highest wealth quantile increases the likelihood of safe waste disposal [5, 18, 22, 23]. The findings of our study are in contrary to those findings. Even though having the highest wealth is important to have easy access to water, sanitation, and hygiene services, it is not a guarantee for safe stool disposal. The highest wealth quantile is necessary but not sufficient to dispose of stools safely. Other factors like knowledge and perception might affect safe stool disposal behavior. That means household income was not associated with safe stool disposal in Gambia. Having high or low wealth quantile is not a precondition to dispose stool safely in Gambia. this implies that during the design of health intervention programs there is a need to consider other factors which may affect stool disposal behavior like knowledge about the importance of safe stool disposal, attitude towards stool disposal and other behavioral as well as socio-cultural issues.

The occupational status of the mothers was significantly associated with safe stool disposal. Mothers who were employed were more likely to dispose of stool safely as compared with unemployed mothers. The finding is inconsistent with a cross-sectional study done in Tigray region of Ethiopia [31]. The possible reason for the discrepancy might be the smaller sample size included in Tigray and the large representative sample used in our study. The other possible reason might be employed mothers have their source of income and they can hire caregivers as well as they can easily access sanitary materials. Therefore, employed mothers would be more likely to dispose of stool safely than mothers who are not employed.

Media exposure was associated with safe child stool disposal. Mothers who were exposed to a media at least once a week were more likely to dispose of stool safely as compared with those who had not exposed. The finding of the study is consistent with what has been reported by a study in in sub-Saharan Africa [21]. Mothers who were exposed to media might get important health information about child waste disposal and its impact on the health of the child as well as the community as a whole, so they might have good knowledge about safe child waste disposal and develop a positive attitude towards the importance of safe child disposal behavior than those mothers who do not exposed. Media has a great impact on behavior change by providing health information to shape the attitude of the community and to promote healthy behavior. Especially Audio-visual education has a great impact on increasing safe child stool practice.

## Strength

Utilization of large sample size and nationally representativeness of DHS data helps to generalize to the population of Gambia. The application of sample weighting to overcome non-proportional sample allocation during the survey. The current study used a large sample size that can help to increase the statistical power and validity of the study.

## Limitations

The main limitation of the study is that since it is a cross-sectional study the temporal relationship between the outcome and independent variables could not be established. The DHS data lacks some variables like knowledge about child stool disposal that may affect their disposal practice. Since the data was collected by using self-reports, there might be a possibility of introducing social desirability bias and recall bias.

## Conclusion

The prevalence of safe child stool disposal was low in Gambia. This might result in diarrheal diseases continued to be an important public health problem, with high morbidity and mortality among children. Safe child stool disposal has a great role in achieving open defecation free communities and reduction of diarrheal diseases. Age of the mother, age of the child, region, media exposure and occupational status of the mother were significantly associated with safe child stool disposal. Household income was not associated with safe stool disposal in Gambia. Public health intervention strategies designed to promote safe child stools disposal need to conduct thorough community assessments to identify community-specific facilitators, needs and barriers. Additionally, public health experts and policy makers should take into consideration the identified factors while designing programs aimed to improve safe child stool disposal practice

## Acknowledgments

We would like to express our gratitude to Demographic Health survey (DHS) program for giving us 2019/20 GDHS data. We also would like to thank study participants.

## Ethics approval and consent to participate

The DHS program conducted the study after getting ethical approval of the Gambia National Ethics Committee. As it is stated in the 2019–20 GDHS report the DHS obtained informed consent from each participant and anonymized the data set during the analysis. Moreover, the data for this study was obtained from the DHS at (http://www.measuredhs.com) after registration and permission had obtained. Since we used secondary data for analysis mentioning the ethical review board and other ethical issues is not applicable. We have obtained permission from the DHS program on reasonable request.

## Author Contributions

**Conceptualization:** Menen Tsegaw, Bezawit Mulat, Kegnie Shitu.

**Data curation:** Menen Tsegaw, Kegnie Shitu.

**Formal analysis:** Menen Tsegaw.

**Investigation:** Menen Tsegaw.

**Methodology:** Menen Tsegaw, Bezawit Mulat, Kegnie Shitu.

**Project administration:** Bezawit Mulat.

**Software:** Menen Tsegaw.

**Supervision:** Bezawit Mulat, Kegnie Shitu.

**Validation:** Menen Tsegaw.

**Visualization:** Menen Tsegaw.

**Writing – original draft:** Menen Tsegaw.

**Writing – review & editing:** Bezawit Mulat, Kegnie Shitu.

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
