## [Decision Letter · Decision Letter 0]

18 Jan 2023

PONE-D-22-33012Safe stool disposal and associated factors among mothers of children under-two age in Gambia: Evidence from Gambia Demographic Health Survey 2019/20PLOS ONE

Dear Dr. Tsegaw,

Thank you for submitting your manuscript to PLOS ONE. After careful consideration, we feel that it has merit but does not fully meet PLOS ONE’s publication criteria as it currently stands. Therefore, we invite you to submit a revised version of the manuscript that addresses the points raised during the review process. Both reviewers provide a number of comments across all sections.   The comments are constructive so should be possible to address, but each ones needs careful attention.

We look forward to receiving your revised manuscript.

Kind regards,

Alison Parker

Academic Editor

PLOS ONE

Journal Requirements:

Reviewers' comments:

Reviewer's Responses to Questions

**Comments to the Author**

1. Is the manuscript technically sound, and do the data support the conclusions?

Reviewer #1: Yes

Reviewer #2: Yes

2. Has the statistical analysis been performed appropriately and rigorously? 

Reviewer #1: Yes

Reviewer #2: Yes

3. Have the authors made all data underlying the findings in their manuscript fully available?

Reviewer #1: Yes

Reviewer #2: Yes

4. Is the manuscript presented in an intelligible fashion and written in standard English?

Reviewer #1: Yes

Reviewer #2: No

5. Review Comments to the Author

Reviewer #1: This is a manuscript about a study in Gambia aimed to assess the prevalence of Safe stool disposal and associated factors among mothers of children under-two age.

1. The main concern of this findings is that Results considered only one option “Those who used toilet/latrine” for Dependent variable assessment which differ their operational definition. The specific type of latrine/toilet use is not mentioned in the results.

The prevalence of safe stool disposal among mothers whose child aged under two years was 56.3% (95%CI: 54.5, 58.1%) which is obtained from the Proportion used toilet/latrine 1697/3011 56.3 %(54.5, 58.1). This is the Dependent variable of the study which is defined as following in Line 62-65:

According to WHO/ UNICEF Joint Monitoring Program, Safe child stool disposal includes

- Flush or pour-flush toilet/latrine to (piped sewer system, septic tank , pit latrine ),

- Ventilated improved pit (VIP) latrine and

- Pit latrine with slab

whereas unsafe stool disposal includes

- pit latrines without a slab or platform,

- hanging latrines and

- bucket latrines

Do they mean all used latrine despite the specific type are labeled as “safe stool disposal”?

2. Authors should also clarify meaning of Media exposure. What does it mean in this context? Operational definition is important in here.

3. Authors also should use referencing manager softwares and correct the citation styles in all the document.

4. To be consistent, in Line 118 (The women’s individual sample weightings …) better if they replace with (The mother’s individual sample weightings …)

Reviewer #2: Little research has been published on safe stool disposal in The Gambia, and this manuscript has some interesting findings which adds to the evidence base.

I have made some comments to improve the paper. Please see the attached PDF for detailed comments.

General:

- There needs to be some revision to the English, which requires editing for correct English

- The tables also need formatting as they are very difficult to view

- I have also suggested where the writing could benefit from expanding to improve what is known in the subject area - at the moment there is not enough comparing/contrasting with other sources in this subject area

- I think you could emphasise more the importance of safe child stool disposal for household health, and also for the health of infants. You might want to search the literature to see if there is any relationship between safe child stool disposal and infant growth/development/malnutrition - you reference this paper (https://www.sciencedirect.com/science/article/abs/pii/S0022347616302438) but I think you should discuss this further in more detail.

- Also, you should refer back to the study context more regularly - why did you choose to focus on The Gambia? Why is safe child stool disposal particularly important in the study context (The Gambia)

Other relevant papers you might want to include and discuss further:

https://bmcpublichealth.biomedcentral.com/articles/10.1186/s12889-022-14309-z

https://www.ncbi.nlm.nih.gov/pmc/articles/PMC7204574/

https://www.mdpi.com/1660-4601/13/3/310

Analysis:

- You should explain in more detail why the sample was weighed - it is not really clear in the paper why this was performed

- I am not sure why you have included p-values for the descriptive data - this should not be included as descriptive analysis is for description only

- Normally we do not include the results from bivariate analyses

- Please be clear in the text which table you are referring to, and clearly label tables with what they show

- It is usual for percentages to be reported to only 1 decimal place

Findings and conclusion:

- You had an interesting finding that safe stool disposal was not associated with the wealth quintile of the household. I think this is a novel finding and you should emphasize this more throughout the paper and try to explain why

- You don't really explain or summarise why it is important that we try to improve safe child stool disposal: you might want to emphasise the importance of safe child disposal for community and household health, and how it might improve infant health also (see my point above under 'general')

- Why is safe child stool disposal particularly important in the study context (The Gambia) - and what might improvement in safe stool disposal mean in terms of improvements in infant health/household health for The Gambia in particular?

I hope the feedback is useful to improve the paper.

6. PLOS authors have the option to publish the peer review history of their article (what does this mean?). If published, this will include your full peer review and any attached files.

Reviewer #1: No

Reviewer #2: No

---

## [Author Response · Author response to Decision Letter 0]

2 Mar 2023

Author’s response to editors and reviewers 

Dear editor in-chief

 Hereby, we resubmit the enclosed revised manuscript ID [PONE-D-22-33012] which is entitled as “Safe stool disposal and associated factors among mothers of children aged under-two years in Gambia: Evidence from Gambia Demographic Health Survey 2019/20" to your journal. 

Reviewer's Responses to Questions

Comments to the Author

1. Is the manuscript technically sound, and do the data support the conclusions?

Reviewer #1: Yes

Reviewer #2: Yes

2. Has the statistical analysis been performed appropriately and rigorously?

Reviewer #1: Yes

Reviewer #2: Yes

3. Have the authors made all data underlying the findings in their manuscript fully available?

Reviewer #1: Yes

Reviewer #2: Yes

4. Is the manuscript presented in an intelligible fashion and written in standard English?

Reviewer #1: Yes

Reviewer #2: No

5. Review Comments to the Author

Reviewer #1: This is a manuscript about a study in Gambia aimed to assess the prevalence of Safe stool disposal and associated factors among mothers of children under-two age.

1. The main concern of this findings is that Results considered only one option “Those who used toilet/latrine” for Dependent variable assessment which differ their operational definition. The specific type of latrine/toilet use is not mentioned in the results.

The prevalence of safe stool disposal among mothers whose child aged under two years was 56.3% (95%CI: 54.5, 58.1%) which is obtained from the Proportion used toilet/latrine 1697/3011 56.3 % (54.5, 58.1). This is the Dependent variable of the study which is defined as following in Line 62-65:

According to WHO/ UNICEF Joint Monitoring Program, Safe child stool disposal includes

- Flush or pour-flush toilet/latrine to (piped sewer system, septic tank , pit latrine ),

- Ventilated improved pit (VIP) latrine and

- Pit latrine with slab

whereas unsafe stool disposal includes

- pit latrines without a slab or platform,

- hanging latrines and

- bucket latrines

Do they mean all used latrine despite the specific type are labeled as “safe stool disposal”?

Response: thank you very much dear for your concern. We have computed the composite variable safe stool disposal by categorizing the listed types of latrines and disposal sites as safe and unsafe according to the WHO/UNICEF criteria. We have computed the outcome variable by recoding “Flush or pour-flush toilet/latrine to (piped sewer system, septic tank, pit latrine), Ventilated improved pit (VIP) latrine and Pit latrine with slab” as safe stool disposal sites which were coded as “latrines”.

2. Authors should also clarify meaning of Media exposure. What does it mean in this context? Operational definition is important in here.

Response: thanks dear. Media exposure was computed as a composite variable using three questions “listens radio, watched TV, and read magazine or newspaper at least once a week”. those who were exposed to at least one form those media for at least once a week were categorized as they had media exposure and labeled as “yes”, where as those who did not have exposure to at least one media were categorized as non-exposed and labeled as “no”. We used the DHS categorization of media exposure as yes/no.

3. Authors also should use referencing manager softwares and correct the citation styles in all the document.

Response: thank you so much dear. We have corrected the errors (see on the highlighted manuscript on page 20-24, line 286-362)

4. To be consistent, in Line 118 (The women’s individual sample weightings …) better if they replace with (The mother’s individual sample weightings …)

Response: we are so grateful dear. We have substituted accordingly (see on the highlighted manuscript on page 6, line 115)

Reviewer #2: Little research has been published on safe stool disposal in The Gambia, and this manuscript has some interesting findings which add to the evidence base. I have made some comments to improve the paper. Please see the attached PDF for detailed comments.

General:

- There needs to be some revision to the English, which requires editing for correct English

- The tables also need formatting as they are very difficult to view

- I have also suggested where the writing could benefit from expanding to improve what is known in the subject area - at the moment there is not enough comparing/contrasting with other sources in this subject area

 I think you could emphasize more the importance of safe child stool disposal for household health, and also for the health of infants. You might want to search the literature to see if there is any relationship between safe child stool disposal and infant growth/development/malnutrition - you reference this paper (https://www.sciencedirect.com/science/article/abs/pii/S0022347616302438) but I think you should discuss this further in more detail.

- Also, you should refer back to the study context more regularly - why did you choose to focus on The Gambia? Why is safe child stool disposal particularly important in the study context (The Gambia)

Response: thank you so much dear for all your constructive comments and questions. We have tried to incorporate all your comments and we are so thankful for that. We have tried to make improvements by including all your comments and questions that you have attached on the automated file. “Diarrheal diseases are continued to be an important public health problem in developing countries, with high morbidity and still significant levels of mortality among children. Since the 1980s, where diarrheal disease control programs were implemented around the world, diarrheal mortality in children under five years of age has been reduced significantly through appropriate case management. However, diarrhea incidence has not changed despite documented progress in water availability and improved sanitation. Therefore, there is a need to better understand the conditions that facilitate diarrhea transmission in less developed countries and to identify and implement more and better interventions designed to interrupt transmission and to decrease the burden of diarrheal diseases.” we have read this quoted finding from a paper “Strategic Report 11 Children's Feces Disposal Practices in Developing Countries and Interventions to Prevent Diarrheal Diseases” and we were inspired to read the stool disposal status in different countries and we have seen the Gambia DHS finding which is the most recent DHS finding “2019/20” then we start exploring about this issue in Gambia. stool disposal has an association with different faecal-oral diseases and identifying the factors affecting safe stool disposal practice has a paramount effect in reducing the burdens of those diseases including the morbidly, mortality, incidence and prevalence. Due to this reason we have selected the topic and the study area. Other relevant papers you might want to include and discuss further:

https://bmcpublichealth.biomedcentral.com/articles/10.1186/s12889-022-14309-z

https://www.ncbi.nlm.nih.gov/pmc/articles/PMC7204574/

https://www.mdpi.com/1660-4601/13/3/310

Response: thank you so much dear for recommending us those important papers. we have included those papers.

Analysis:

- You should explain in more detail why the sample was weighed - it is not really clear in the paper why this was performed

Response: thanks dear. The mother’s individual sample weightings were used in the estimation to provide nationally representative results and to overcome non-proportional sample allocation during the survey. 

- I am not sure why you have included p-values for the descriptive data - this should not be included as descriptive analysis is for description only

Response: thanks dear. We have removed the p-value. (See on the manuscript on page 7, line 129)

- Normally we do not include the results from bivariate analyses

- Please be clear in the text which table you are referring to, and clearly label tables with what they show

Response: thank you so much dear. we have corrected editorial errors through out all tables and texts. 

- It is usual for percentages to be reported to only 1 decimal place

Response: thanks dear. We have changed percentages into one decimal place (See on the manuscript on page 7-10, line 129)

Findings and conclusion:

- You had an interesting finding that safe stool disposal was not associated with the wealth quintile of the household. I think this is a novel finding and you should emphasize this more throughout the paper and try to explain why

Response: thank you dear. We have tried to add some explanations about this finding (see on the manuscript on page 16-17, line 211-215 and page 18, line 251).

- You don't really explain or summarise why it is important that we try to improve safe child stool disposal: you might want to emphasise the importance of safe child disposal for community and household health, and how it might improve infant health also (see my point above under 'general')

Response: thanks a lot dear. We have tried to include some concepts regarding importance of safe child disposal in the conclusion part finding (see on the manuscript on page 18, line 246-249)

- Why is safe child stool disposal particularly important in the study context (The Gambia) - and what might improvement in safe stool disposal mean in terms of improvements in infant health/household health for The Gambia in particular?

Response: we are so grateful dear. Safe stool disposal has a great benefit in reducing the transmission, incidence, prevalence, morbidity, and mortality of faecal-oral diseases including diarrhea in developing countries including.

I hope the feedback is useful to improve the paper.

Response: we cannot thank you enough dear for your time, constructive comments and interesting questions. We hope that we have addressed your concerns very well and your points help us to made substantial changes in our manuscript.

Dear reviewers, 

We have read the comments carefully and we were able to implement all of them. While reading the manuscript critically, we spotted English grammar errors, which we have corrected too. We hope that our revision will be felt like an improvement. We certainly feel this manuscript has improved thanks to you (reviewers) for your suggestions.

6. PLOS authors have the option to publish the peer review history of their article (what does this mean?). If published, this will include your full peer review and any attached files.

Do you want your identity to be public for this peer review? For information about this choice, including consent withdrawal, please see our Privacy Policy.

Reviewer #1: No

Reviewer #2: No

---

## [Decision Letter · Decision Letter 1]

13 Apr 2023

Safe stool disposal and associated factors among mothers of children aged under-two years in Gambia: Evidence from Gambia Demographic Health Survey 2019/20

PONE-D-22-33012R1

Dear Dr. Tsegaw,

We’re pleased to inform you that your manuscript has been judged scientifically suitable for publication and will be formally accepted for publication once it meets all outstanding technical requirements.

Kind regards,

Alison Parker

Academic Editor

PLOS ONE

Additional Editor Comments (optional):

Reviewers' comments:

Reviewer's Responses to Questions

**Comments to the Author**

1. If the authors have adequately addressed your comments raised in a previous round of review and you feel that this manuscript is now acceptable for publication, you may indicate that here to bypass the “Comments to the Author” section, enter your conflict of interest statement in the “Confidential to Editor” section, and submit your "Accept" recommendation.

Reviewer #2: All comments have been addressed

2. Is the manuscript technically sound, and do the data support the conclusions?

Reviewer #2: Yes

3. Has the statistical analysis been performed appropriately and rigorously? 

Reviewer #2: Yes

4. Have the authors made all data underlying the findings in their manuscript fully available?

Reviewer #2: Yes

5. Is the manuscript presented in an intelligible fashion and written in standard English?

Reviewer #2: (No Response)

6. Review Comments to the Author

Reviewer #2: Well done on the amends to this paper. The paper seems much improved and I am glad the feedback was useful.

7. PLOS authors have the option to publish the peer review history of their article (what does this mean?). If published, this will include your full peer review and any attached files.

Reviewer #2: No

---

## [Editor Report · Acceptance letter]

19 Apr 2023

PONE-D-22-33012R1 

Safe stool disposal and associated factors among mothers of children aged under-two years in Gambia: Evidence from Gambia Demographic Health Survey 2019/20 

Dear Dr. Tsegaw:

I'm pleased to inform you that your manuscript has been deemed suitable for publication in PLOS ONE. Congratulations! Your manuscript is now with our production department. 

Kind regards, 

on behalf of

Dr. Alison Parker 

Academic Editor

PLOS ONE